# Transcriptome Profiling of Stem-Differentiating Xylem in Response to Abiotic Stresses Based on Hybrid Sequencing in *Cunninghamia lanceolata*

**DOI:** 10.3390/ijms232213986

**Published:** 2022-11-12

**Authors:** Wentao Wei, Huiyuan Wang, Xuqing Liu, Wenjing Kou, Ziqi Liu, Huihui Wang, Yongkang Yang, Liangzhen Zhao, Hangxiao Zhang, Bo Liu, Xiangqing Ma, Lianfeng Gu

**Affiliations:** 1College of Forestry, Fujian Agriculture and Forestry University, Fuzhou 350002, China; 2Basic Forestry and Proteomics Research Center, College of Forestry, School of Future Technology, Fujian Agriculture and Forestry University, Fuzhou 350002, China

**Keywords:** *Cunninghamia lanceolate*, stem-differentiating xylem, abiotic stress, nanopore long-read sequencing, transcription factors

## Abstract

*Cunninghamia lanceolata* (*C. lanceolata*) belongs to Gymnospermae, which are fast-growing and have desirable wood properties. However, *C. lanceolata*’s stress resistance is little understood. To unravel the physiological and molecular regulation mechanisms under environmental stresses in the typical gymnosperm species of *C. lanceolata*, three-year-old plants were exposed to simulated drought stress (polyethylene glycol 8000), salicylic acid, and cold treatment at 4 °C for 8 h, 32 h, and 56 h, respectively. Regarding the physiological traits, we observed a decreased protein content and increased peroxidase upon salicylic acid and polyethylene glycol treatment. Superoxide dismutase activity either decreased or increased at first and then returned to normal under the stresses. Regarding the molecular regulation, we used both nanopore direct RNA sequencing and short-read sequencing to reveal a total of 5646 differentially expressed genes in response to different stresses, of which most had functions in lignin catabolism, pectin catabolism, and xylan metabolism, indicating that the development of stem-differentiating xylem was affected upon stress treatment. Finally, we identified a total of 51 AP2/ERF, 29 NAC, and 37 WRKY transcript factors in *C. lanceolata*. The expression of most of the NAC TFs increased under cold stress, and the expression of most of the WRKY TFs increased under cold and SA stress. These results revealed the transcriptomics responses in *C. lanceolata* to short-term stresses under this study’s experimental conditions and provide preliminary clues about stem-differentiating xylem changes associated with different stresses.

## 1. Introduction

Plants, as sessile organisms, experience constant environmental stresses throughout their life cycle [1]. Exposure to different stresses may lead to the development of sophisticated mechanisms due to physiological changes and gene transcription allowing the plant to survive and cope with the stresses [2]. At the beginning of the stress response, stress perception occurs on the cell surface or cell membrane [3]; then, a primary physiological response is stimulated during early stress, such as protein content [4]. Abiotic stress disrupts the metabolic balance of cells, resulting in excessive ROS, which attacks biomembranes [5], and then directly affects the malondialdehyde (MDA) content and superoxide dismutase (SOD) and peroxidase (POD) activities [6,7,8].

Extreme drought and temperature conditions have long been known to have important effects on plants [9]. Drought and chemical treatments, such as *β*-aminobutyric acid and salicylic acid (SA), can induce plants to generate higher expression of stress-response genes to protect against future abiotic stress [10,11,12,13]. For example, *AtLTI30*, *NFYA5*, *AtNAC72 (RD26)*, *AtNAC19*, and *AtNAC55* are important drought-response genes in *Arabidopsis* [14,15,16,17]. *OsNAC14* and *OsTPP3* enhance drought tolerance in *Oryza sativa* [18,19]. Overexpression of *PtrNAC006*, *PtrNAC007*, and *PtrNAC120* results in obvious drought tolerance in *Populus trichocarpa* [20]. SA, an endogenous signal molecule, involved in plant growth and development, also participates in the regulation of stress resistance in plants [21,22,23,24,25]. *OsNPR1* and *OsWRKY45* play complementary roles in the SA pathway in response to environmental stresses [26]. In addition to drought and SA treatment, ambient temperature also regulates growth, flowering, and morphology [27,28]. Cold tolerance can be increased by a prior exposure to a low, nonfreezing temperature (known as cold acclimation), such as the *CBF* genes that regulate 414 cold-responsive (*COR*) genes [29]. *IbCBF3* enhances cold and drought tolerance in potato, and *OsSMP1* increases cold tolerance in rice [30,31].

Transcription factors (TFs) play a pivotal role in plant signaling pathways for plant responses to biotic and abiotic stress, and also play an important role in responses to internal signals of interacting networks among developmental processes [32,33,34]. Land plants present multiple kinds of TFs such as, *AP2/ERF*, *ARF*, *bHLH*, *bZIP*, *C2H2*, *Dof*, *HSF*, *MYB*, *NAC*, and *WRKY* [35]. AP2/ERF family factors are mainly classified into four major subfamilies: DREB (dehydration-responsive element-binding) proteins, ERF (ethylene-responsive element-binding) proteins, AP2 (APETALA2) and RAV (related to ABI3/VP), and soloists (a few unclassified factors) [34]. The NAC (NAM, CUC, and ATAF) TFs are characterized by a highly conserved DNA-binding NAC domain (~150 amino acids) in the N-terminal region [32]. The first reported NAC proteins are NAM from *Petunia hybrida* [36] and the Ataf1/2 and Cuc2 proteins in *Arabidopsis* [37]. The WRKY TF family of genes comprises one or two conservative WRKY domains, a WRKYGQK motif at the N-terminus, and a C_2_-H_2_ or C_2_-HC zinc finger structural domain in the C-terminus [38]. According to the number of WRKY domains and the characteristics of the zinc finger domains, WRKY is mainly divided into three categories: I (two WRKY DBDs), II (a single DBD with different C2–H2 zinc fingers), and III (a single DBD with C2–HC zinc finger [38]; additionally, some reports have revealed a fourth category (an incomplete WRKY domain without zinc fingers) [33]. TFs participate not only in stress responses but also in the development of wood formation. For example, *PtrMYB3* and *PtrMYB20* of MYB TFs were found to be involved in the biosynthesis of secondary xylem [39,40]. *PtaERF1* has a potential role in phloem development [41]. Several TFs, including MYB genes, AP2/ERF genes, WRKY genes, and NAC genes, display significantly up or downregulated expression levels during the development of wood formation [42].

Our previous study showed that *MYB2*, *MYB21*, and *MYB092* are differentially expressed upon compression stress in *Cunninghamia lanceolata* (Lamb.) Hook (*C. lanceolata*) [43]. *C. lanceolata*, as endemic and evergreen coniferous species, is one of the most important artificial timber forest species in southern China because of its high timber quality and fast growth [43,44]. *C. lanceolata* is gradually suffering from increased stress, including temperature [45], drought stress [46], and autotoxicity caused by allelopathic toxins such as phenolic acids [47]. However, the physiological and molecular regulation mechanisms under environmental stress are less reported in the typical gymnosperm species of *C. lanceolata*. We hypothesized that *C. lanceolata*, as a gymnosperm plant, has certain tolerance mechanics in response to different stresses, which could be applied to improve the stress tolerance for plants. Therefore, we investigated three-year-old *C. lanceolata* trees that were exposed to simulated drought stress (20% (*w*/*v*) PEG-8000), 5 mM SA, and 4 °C cold treatment in order to reveal the potential response patterns in this gymnosperm species. In this work, we also revealed the changes in *AP2/ERF*, *NAC*, and *WRKY* TFs upon stress treatment. These results provide basic clues regarding how to enhance stress tolerance in *C. lanceolata*.

## 2. Results

### 2.1. Physiological Responses of C. lanceolata under Various Stresses

Simulated drought stress (PEG), SA treatment, and low temperatures stimulated different changes in the soluble protein content, MDA, POD activity, and SOD activity across four time points (0, 8, 32, and 56 h) (Figure 1). The PEG treatment decreased the soluble protein content and SOD activity and increased POD activity after eight hours. Under the SA treatment, the soluble protein content and SOD gradually decreased, and POD increased at 32 h. Short-term cold treatment did not cause any significant effects on the soluble protein content, MDA, or POD, and only significantly increased SOD in the 8 h cold treatment. In summary, drought stress was found to stimulate significant changes in the protein content, SOD, and POD. However, the SA and cold treatment only caused subtle changes in the protein content, SOD, MDA, and POD, which then returned to normal levels.

### 2.2. Constructing Full-Length UniGene Clusters Based on Nanopore Long Reads

We collected the secondary xylem materials upon stress treatment including PEG, SA, and low-temperature treatments at 0 h, 8 h, 32 h, and 56 h (Figure 2). Three biological repeats were taken for each treatment for RNA-seq. As *C. lanceolata* has a large genome size, its reference sequence and gene annotation are not available. In order to obtain a high-quality annotated sequence of the reference transcripts, we performed nanopore DRS to obtain a high-quality long transcript reference sequence. Using nanopore direct RNA sequencing, 1,031,529 reads were generated. In order to ensure the accuracy of the sequence, we used RNA-seq reads from Illumina transcriptome data to correct the long reads before UniGene clustering. Finally, 15,348 genes including 62,558 transcript isoforms were obtained. Due to the depth of RNA direct sequencing, some low-abundance genes might have been omitted. The same samples of this project were deeply sequenced using Illumina RNA-seq. Finally, we obtained a total of 22,026 genes as the reference transcripts.

### 2.3. Identification of Differentially Expressed Genes under Stress Treatments

Principal component analysis (PCA) showed that three biological repeats of each treatment were clustered near each other, which reflected the reliability of the high-throughput sequencing dataset (Figure 3A). The cold treatment dataset was far from the other samples, indicating that the cold treatment produced more obvious changes than the PEG and SA treatments. In order to obtain the DEGs under different stresses, we compared different stress treatments with the control. The genes with standardized expression changes greater than 2 and *p*-values less than 0.05 were considered as DEGs. Under the cold treatment, we identified 1776, 1531, and 1626 DEGs at 8 h, 32 h, and 56 h, respectively (Figure 3B, Appendix A). Under the PEG treatment, 1653, 1524, and 1261 DEGs were identified at 8 h, 32 h, and 56 h, respectively. Under the SA treatment, 1309, 1243, and 1033 DEGs were identified at 8 h, 32 h, and 56 h, respectively. In this study, TFs were also involved in transcription regulation under different stresses in *C. lanceolata*. Among all the DEGs, there were 264 TF genes in 47 families (Figure 3B), including the APETALA2/ethylene-responsive factor (AP2/ERF) family, the NAC family, and the WRKY family.

A total of 5646 DEGs were found from all the stress treatments. Noticeably, the GO analysis showed that these DEGs were enriched in the pathway of lignin catabolism, pyrimidine metabolism, and xyloglucan metabolism (Figure 3C–E).

We further analyzed the dynamic transcriptomes of the four treatment points (0 h, 8 h, 32 h, and 56 h) under three different stress treatments. According to the co-expression profiles of the differentially expressed genes under the different stresses and treatment points, they could be divided into 330 clusters. Among these, cluster 2 was the category with the maximum number of genes, which contained 617 DEGs, showing a rapid up-regulated expression at 8 h of cold treatment (Figure 4A, Appendix A), such as the MYB family gene *TCONS_00011519*. The GO enrichment analysis revealed that the genes in cluster 2 were mainly in the phosphoprotein phosphatase and abscisic acid (ABA)-mediated signal pathway (Figure 4B). The genes of cluster 254 were obviously upregulated at all points of the cold treatment, but the changes were not obvious under the other types of stress (Figure 4C; Appendix A). The GO terms in cluster 254 were mainly enriched in biological processes regulated by cell metabolism, the regulation of transcription factor activity, and circadian rhythm (Figure 4D). The expression profile of cluster 7 increased rapidly at 8 h of SA treatment, but did not change significantly in the other treatments (Figure 4E, Appendix A), which indicated that these genes were associated with SA response. The GO terms in cluster 7 were mainly enriched in biological processes of transport genes, such as triose phosphate transport, phosphoglycerate transport, hexose transport, and glucose transport (Figure 4F). In contrast to the above clusters, the genes in cluster 58 were downregulated under all the treatments (Figure 4G, Appendix A), while the GO terms in cluster 58 were mainly enriched in biological processes related to cell recognition and communication (Figure 4H). In summary, these DEGs might be associated with stress response, which provides preliminary clues for further investigation of the stress tolerance mechanisms.

### 2.4. Phylogenetic and Expression Analysis of the AP2/ERF Family

We identified 51 AP2/ERF members from the UniGene reference transcriptome (Figure 5A). Using homology alignment, we found one member of the A-1 subfamily, one member of the A-2 subfamily, four members of the A-4 subfamily, one member of A-5 subfamily, and five members of A-6 subfamily in *C. lanceolata* (Figure 5A). In *C. lanceolata*, we found that there was one gene (ed224c4a-e61f-4107-8337-d2e4170696f3) that was homologous with CBF genes (Figure 5A). The expression level of *TINY2* from the A-4 subfamily and *RAP2-3* from the A-5 subfamily decreased under the cold, PEG, and SA stress treatments (Figure 5B). *RAP2.10* belonging to the A-5 subfamily was downregulated under all the treatments, which was validated using RT-qPCR (Figure 5C). Most of the AP2/ERF genes with differentially expressed changes were derived from the ERF subfamily. For example, *ERF1-like*, *ERF003-like*, *ERF071-like-2*, and *ERF2-like* were all downregulated under the different stress treatments. *Arabidopsis thaliana ABR1* is a member of the ERF B-4 subfamily. In this study, the expression level of *ABR1* increased significantly under cold treatment, but decreased significantly under PEG and SA treatment (32 h and 56 h), which was consistent with the RT-qPCR verification (Figure 5D).

### 2.5. Phylogenetic and Expression Analysis of NAC Gene Family Members

NAC family members play important roles in plant resistance [17]. The gene family analysis revealed 29 members of the NAC family in *C. lanceolata* (Figure 5E). Most of the NAC genes were significantly upregulated under the cold treatment (Figure 5F), such as *TCONS_00012497*, *TCONS_00019374*, and *TCONS_00030718*, which were found to be homologous with *ATAF1*. *ATAF1* is involved in a broad range of biotic and abiotic stresses [48,49], which suggests that these genes might be associated with stress response. The RT-qPCR verification of *TCONS_00064908* and *TCONS_00017872* confirmed their significantly increased expression under cold stress and other stresses (Figure 5G,H).

### 2.6. Phylogenetic and Expression Analysis of WRKY Family Members

As one of the largest families of transcriptional regulators, WRKY TFs can act as activators or repressors in some homo- and heterodimer combinations [50]. In this study, we identified 37 WRKY family members in *C. lanceolata* (Figure 5I). The evolutionary analysis revealed that *TCONS_00030319* was clustered in the same branch with *WRKY33, WRKY34*, *WRKY26*, *WRKY2*, and *WRKY20*. *TCONS_00030623* and *TCONS_00032040* presented similar homology with *WRKY42*, *WRKY47*, and *WRKY6*. *TCONS_00024961* was clustered in the same branch with *Arabidopsis thaliana WRKY14* and *WRKY35*. From the heatmap analysis, most of the WRKY TFs were significantly increased under the cold and SA treatments (Figure 5J). The RT-qPCR verification of *TCONS_00032040* and *TCONS_00016922* also validated their increased expression under cold and SA stresses (Figure 5K,L).

## 3. Discussion

In this study, we found that drought stress could stimulate significant changes in protein content, SOD, and POD. The SA and cold treatments only caused subtle changes in protein content, SOD, MDA, and POD, which then returned to normal levels. A total of 5646 DEGs were found under all the stress treatments in *C. lanceolata*. According to the co-expression profiles of the differentially expressed genes under the different stresses and treatment points, they could be divided into 330 clusters. In particular, we identified 264 TF genes in 47 families among DEGs, including the APETALA2/ethylene-responsive factor (AP2/ERF) family, the NAC family, and the WRKY family.

### 3.1. Physiological Response to Drought, Cold, and SA Treatment

Plants constantly monitor and cope with fluctuating environments, such as drought, cold, salinity stress, and so on [51]. Drought contributed to the increase in soluble protein and SOD activity and the decrease in POD activity but had no obvious effect on MDA content in highland barley [52]. In this study, the PEG treatment decreased the soluble protein content and SOD, and increased POD after eight hours (Figure 1). This result is somewhat from the results obtained from *Larix principis-rupprechtii* seedlings [53] and cotton [54] under drought, which have might been caused due to the differences in these species.

*Brassica napus* [55] and highland barley [52] have their own specific response characteristics of physiological tolerance in different climatic regions with different temperatures. In *C. lanceolata*, the short-term cold treatment did not cause any significant changes in the soluble protein content, MDA, and POD, and only significantly increased SOD at 8 h of cold treatment (Figure 1). However, long-term cold treatment is necessary to reveal the distribution of *C. lanceolata* in China.

SA and cold treatment only caused short-time changes, which then returned to normal levels in *C. lanceolata*. Noticeably, the changes in MDA content were not significant under these three stresses (Figure 1B). Different plants have developed their own specific stress adaptation strategies through long-term evolution [7,56]. *C. lanceolata*, as a gymnosperm woody forest species, might have developed short-term tolerance to prevent severe oxidative damage through a self-protecting strategy (Figure 6). However, it is worth investigating long-term stress in further research.

### 3.2. Differentially Expressed Patterns of Secondary Xylem in Response to Drought, Cold, and SA Treatments

In addition to PEG, cold, and SA-induced physiological responses, transcriptional regulation of genes was also observed in the secondary xylem of *C. lanceolata* (Figure 3B and Figure 6). In this study, we obtained a total of 5646 DEGs under all the stress treatments in the secondary xylem. These DEGs were enriched in the pathway of lignin catabolism, pyrimidine metabolism, and xyloglucan metabolism (Figure 3C–E), indicating that stress might affect the biosynthesis of the secondary xylem of *C. lanceolata*, which was consistent with our previous study on *Populus trichocarpa* [57].

Conifers with strong adaptability to harsh environments generally have large distribution areas covering multiple climate zones, dominating boreal and cold climates in the Northern Hemisphere [58,59]. The cold treatment presented more obvious changes in transcriptional regulation than the PEG and SA treatments (Figure 3B). In *Pinus tabuliformis*, several AP2/ERF members such as *PtDREB1* and *PtDREB2* were highly cold responsive [60]. Therefore, we speculate that *C. lanceolata* has developed a transcriptional mechanism by which to rapidly respond to short-term cold stress. However, it will be worth investigating how long-term stress treatment or periodical abiotic stresses affect the “stress memory” that primes the plant for faster and stronger responses upon subsequent stress [2].

### 3.3. TF Expression under Drought, Cold, and SA Stress

In this study, we identified 264 TF genes in 47 families, including the AP2/ERF family, the NAC family, and the WRKY family, respectively. Among these, 51 AP2/ERF members, 29 NAC members of the NAC family, and 37 WRKY family members were identified in *C. lanceolata* (Figure 5 and Figure 6). There was a total of 188 TFs/TRs that significantly responded to various environmental stresses, including cold, moderate and progressive drought, and other stresses, in *Pinus tabuliformis* [60]. *PtDREB1* and *PtDREB2* are important cold-responsive AP2/ERF members in *Pinus tabuliformis* [60]. Temperate plants need cold acclimation to acquire freezing tolerance after a period of low, nonfreezing temperatures; for example, the AP2/ERF TFs were found to directly activate cold-responsive (*COR*) genes in *Arabidopsis* [29,52,54]. In *C. lanceolata*, most AP2/ERF TFs belong to the ERF subfamily and the DREB subfamily (Figure 5A), and the ERF subfamily members presented more changes under different stresses (Figure 5A,B). Although there was no CBF subfamilies that were critical for cold acclimation in *Pinus tabuliformis* and other conifer genomes [60], we predicted that one gene (ed224c4a-e61f-4107-8337-d2e4170696f3) was homologous with CBF genes (Figure 5A).

AP2/ERF, NAC, and several WRKY TFs all participate in “survival of the fittest” [61]. In our study, most of the NAC TFs and WRKY TFs, plus some key AP2/ERF members, were significantly increased under cold stress (Figure 5B,F,J). It will be interesting to investigate the cooperative network in AP2/ERF, NAC, and WRKY members in response to different stresses. Many studies have reported that NAC, AP2/ERF, and WRKY participate in the lignin synthesis-related pathway [41,42,62,63,64]. Thus, investigations on the interplay between growth and multiple types of stress are required in the future to reveal how *C. lanceolata* regulates key responsive TFs to balance its growth and stress responses. In particular, we found differentially expressed TFs. It is possible that microRNAs also have a role in this process since TFs comprise most of the miRNA target genes [65]. Small RNA libraries and degradome libraries [66] can address this possibility in the future.

## 4. Materials and Methods

### 4.1. Stress Treatment

*C. lanceolata* trees were grown in a greenhouse at 25 °C with 16 h light/8 h dark cycles in Fuzhou, China. Three-year-old *C. lanceolata* were divided into four groups: (1) control (untreated plants), (2) simulated drought stress treatment with 20% (*w*/*v*) PEG-8000, (3) exogenous 5 mM salicylic acid (SA), and (4) cold treatment in a growth chamber at 4 °C. In order to avoid the effect of circadian rhythms on gene expression, we collected materials at 24 h intervals. Thus, all the plants under the different stresses were treated at ~8:00 a.m., and we harvested secondary xylem and leaves after 0 h, 8 h, 32 h, and 56 h. In total, we selected nine individual plants as one biological repeat for one time interval for each treatment. In total, three biological repeats were collected. The above materials were immediately frozen in liquid nitrogen and stored at −80 °C until use for subsequent high-throughput sequencing and real-time quantitative PCR (RT-qPCR). Three biological repeats were used for sequencing and qPCR.

### 4.2. Determination of MDA, Protein Content, SOD, and POD

The frozen leaves (0.1g) with three biological repeats were crushed into a fine powder using a tissue crusher (QIAGEN TissueLyser Ⅱ, Dusseldorf, Germany), and then suspended in 0.9 mL PBS solution (0.1 mol/L, pH 7.0) in a volume ratio of 1:9. The homogenate was centrifuged at 4 °C with 4000 rpm for 10 min. The clear supernatant was used for the determination of MDA, protein content, SOD, and POD using commercial assay kits (Nanjing Jiancheng Bioengineering Institute, Nanjing, China). The protein contents were detected using a Thermo Fisher Multiskan™ FC microplate photometer and determined using the total protein assay kit (with the standard BCA method) (Nanjing Jiancheng, A045-4); the protein concentration was calculated by measuring the absorbance in 562 nm. We took the protein content changes as physiological traits since the protein content was used in the determination of SOD and POD activity as described in the total superoxide dismutase (T-SOD) assay kit (hydroxylamine method) (Nanjing Jiancheng, A001-1) and the peroxidase assay kit (A084-3, Plant). The activity of MDA was determined using an MDA assay kit (TBA method, A003-1) [67]. The activity of SOD was determined using a total superoxide dismutase (T-SOD) assay kit (hydroxylamine method) (Nanjing Jiancheng, A001-1) [68]. The POD activity was measured using a peroxidase assay kit (A084-3, Plant), measuring the change in absorbance at 420 nm. Three biological repeats of the physiological traits were measured for each stress condition.

### 4.3. Illumina Short-Read Library Preparation and Sequencing

Total RNAs were extracted from secondary xylem materials treated with SA, PEG, and cold stress at four time points (0 h, 8 h, 32 h, and 56 h) using the RNAprep Pure Plant Kit (Tiangen, DP441, Beijing, China). RNA quality was determined using a NanoDrop 2000c UV–Vis spectrophotometer (Thermo Fisher, Waltham, MA, USA) followed by 1% agarose gel electrophoresis. High-quality total RNAs with RNA integrity number (RIN) values higher than 8 were used in subsequent Illumina RNA-seq, and RT-qPCR. In this study, we constructed strand-specific RNA-Seq libraries including the control and the PEG, SA, and low-temperature treatments for 0 h, 8 h, 32 h, and 56 h. The RNA-Seq was constructed using the deoxyuridine triphosphate (dUTP) method using Illumina Adapter Sequences (5’ AATGATACGGCGACCACCGA GATCTACACIIIIIIIIAC ACTCTTTCCCTACACGACGC TCTTCCGATCT (-) AGATCGGAAGAGCACACGTC TGAACTCCAGTCACIIIIII IIATCTCGTATGCCGTCTTCTG CTTG 3’) and the Illumina TruSeq Stranded mRNA kits. Three biological repeats were sequenced for each treatment based on the Illumina HiSeq platform (Illumina, San Diego, CA, USA) using the 150 bp paired-end strategy. 

### 4.4. RNA-Seq Analysis Methods

The low-quality reads were removed using the ht2-filter from HTQC [69]. Then, we used RNA-Seq by Expectation Maximization (RSEM) [70] and empirical analysis of DGE in R (edgeR) [71] to calculate the gene expression and the differentially expressed genes under the cold, PEG, and SA stress treatments. In order to assemble unigenes, we generated Illumina-based reference transcripts using cufflinks (-F 0.5--min-frags-per-transfrag 30--min-intron-length 30) and cuffmerge with the default parameters [72]. Using differentially expressed genes as input files, a co-expression analysis of the differentially expressed genes (DEGs) was performed using the default parameters of the Dirichlet process Gaussian process mixture model software [73]. DPGP (https://github.com/PrincetonUniversity/DP_GP_cluster (accessed on 9 September 2019)) performs very well in clustering gene expression time-series data using an infinite Gaussian process mixture model. After we obtained the DEGs, we extracted the FPKM values of the DEGs. Then, we used the FPKM values as inputs for the DPGP to cluster with the default parameters so as to identify similar expression disjoint clusters.

### 4.5. Nanopore Direct RNA Sequencing and Bioinformatics Analysis

The genome of *C. lanceolata* was not available due to its large size. Thus, we also used long-read sequencing based on nanopore direct RNA-seq (DRS) to generate a high quantity of unigenes and to improve the integrity of the unigenes. The RNA quality was determined using a NanoDrop 2000c UV–Vis spectrophotometer (Thermo Fisher, Waltham, MA, USA) followed by 1% agarose gel electrophoresis, and the high-quality total RNA with RIN values higher than 8 was used. The total RNAs of the secondary xylem from different stages and different treatments were mixed into one sample for DRS, and mRNA was isolated using the Dynabeads mRNA Purification Kit (AMBION, CAT#61006). The DRS library was built using the SQK-RNA002 Direct RNA Sequencing Kit (Oxford Nanopore Technologies). The constructed library was loaded onto the R.9.4.1 FlowCell (FLO-MIN106) for sequencing in a MinION MK 1B sequencer running for 48 h. After sequencing, the electronic raw signals were identified using Guppy (version 2.3.1) with the default parameters. The transcriptome data generated by DRS were corrected by lordec-correct (-k 21 -s 3) [74] using RNA-seq reads to obtain more accurate long transcripts. Finally, reference transcript sequences were generated using CD-HIT software [75] to remove redundancy.

### 4.6. Gene Ontology Enrichment Analysis

In this study, the gene ontology (GO) terms of each unigene were obtained using BLAST2GO [76]. The GO enrichment analysis was implemented using the clusterProfiler package [77]. Directed graphs of the GO were obtained using Cytoscape’s BINGO plug-in [78] with the default parameters.

### 4.7. Evolution Analysis of Transcription Factors

A total of 960 TFs were identified using the iTAK [79] database with the default parameters. The phylogenetic tree was constructed using MEGAX [80] with the neighbor-joining (NJ) method with 1000 bootstrap values.

### 4.8. RT-qPCR Verification

RT-qPCR was used to verify the gene expression patterns of the AP2 family, the NAC family, and the WRKY family under drought, low-temperature, and SA treatments. Premier 5 was used to design the primers [81]. All the information of each primer pair can be found in the attachment (Appendix A), and the RNA-Seq results were verified using the SYBR^®^ Green premix ProTaq HS qPCR Kit (Accurate Biotechnology (Human) Co., Ltd., Changsha, China). The instrument was the Applied Biosystems QuantStudio™ 6 Flex Real-Time PCR System (Thermo Fisher, Waltham, MA, USA). Ribosomal 5S (TCONS_00059130) was used as a positive control for the validation of each gene. Three biological repeats were measured for each stress condition.

## 5. Conclusions

In this study, we investigated the changes in physiological markers in *C. lanceolata* upon drought, SA, and cold treatment at different time points (Figure 6). Moreover, we discovered a total of 5646 DEGs in response to multiple stresses by both nanopore DRS sequencing and short-read sequencing. The GO terms related to lignin catabolism and xylan metabolism were enriched. In particular, we identified multiple TFs in response to short-term stress, which provided candidate genes for the improvement of stress resistance of *C. lanceolata*. These results revealed the physiological and transcriptomics responses to short-term stresses in *C. lanceolata* and provide preliminary clues about SDX changes associated with different types of stress. Additionally, our work also provides a molecular direction for future studies regarding the screening of resistance genes in typical gymnosperm species.

## Figures and Tables

**Figure 1 ijms-23-13986-f001:**
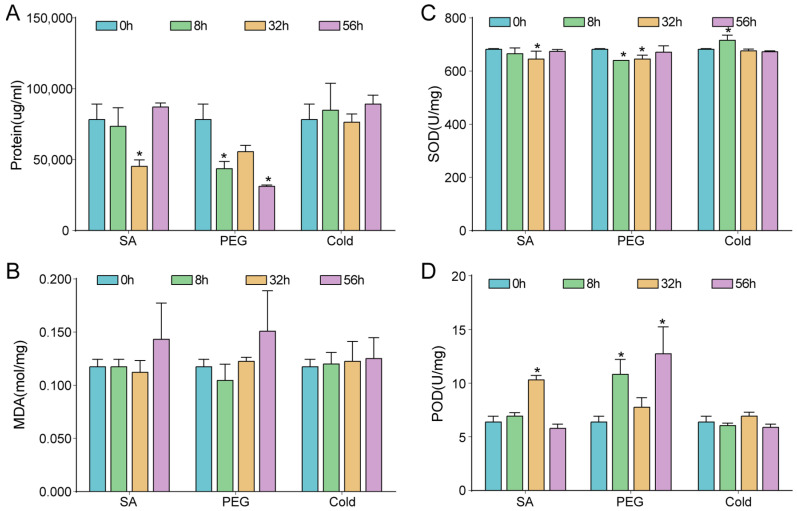
Physiological trait measurements after PEG, SA, and low-temperature treatment for 0 h, 8 h, 32 h, and 56 h in *C. lanceolata*. (**A**) the amount of soluble protein content in different treatments; (**B**) the amount of MDA content in different treatments; (**C**) the amount of SOD activity in different treatments; (**D**) the amount of POD activity in different treatments. (*, *p* < 0.05, *t*-test).

**Figure 2 ijms-23-13986-f002:**
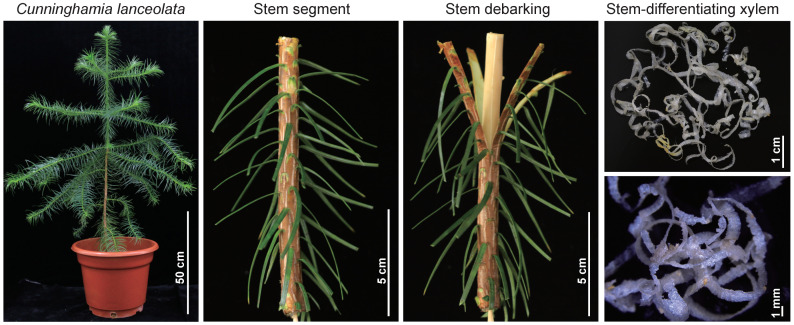
Schematic drawing of secondary xylem of *C. lanceolata*. After PEG, SA, and low-temperature treatments for 0 h, 8 h, 32 h, and 56 h, the stem-differentiating xylem was scraped with a blade.

**Figure 3 ijms-23-13986-f003:**
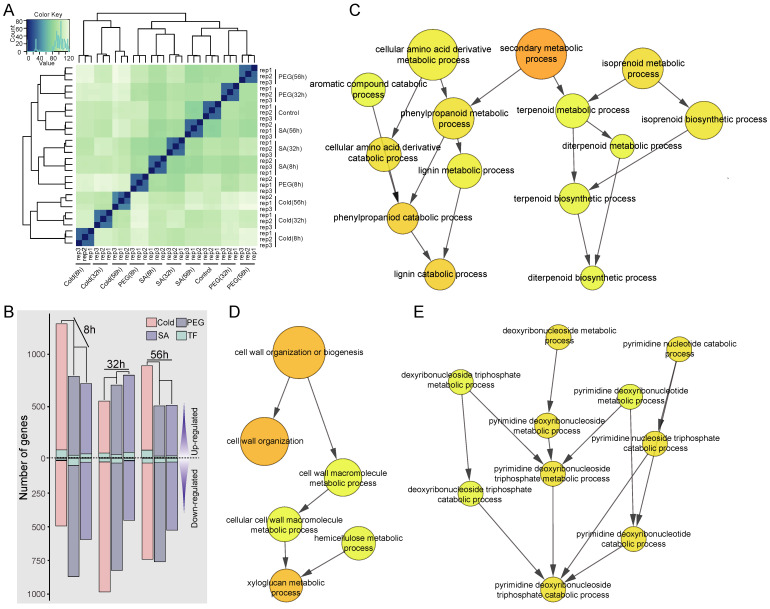
Enrichment analysis of differentially expressed genes and gene ontology under different stress treatments. (**A**) principal component analysis for RNA-Seqs; (**B**) pink, gray, and purple represent differentially expressed genes under cold, PEG, and SA treatments, respectively; blue color represents TFs; (**C**) Differentially expressed genes were enriched in lignin catabolism; (**D**) Differentially expressed genes were enriched in xyloglucan metabolism; (**E**) Differentially expressed genes were enriched pyrimidine metabolism.

**Figure 4 ijms-23-13986-f004:**
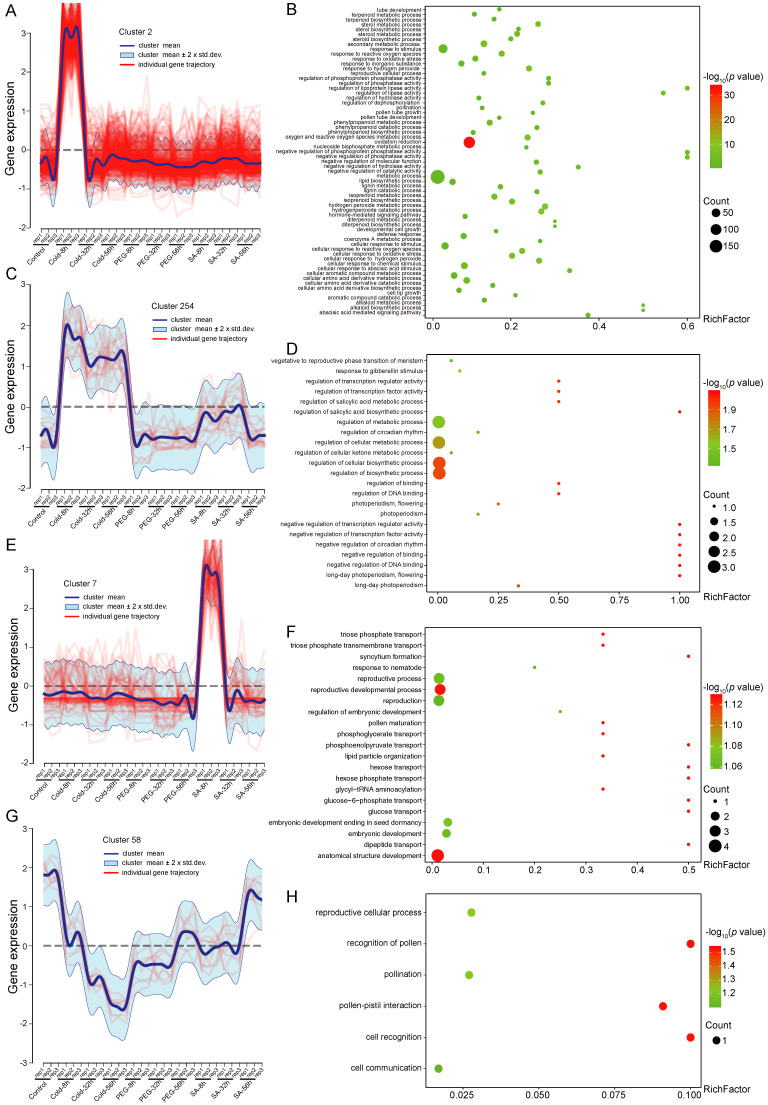
Co-expression analysis of differential genes. (**A**) gene co-expression patterns of cluster 2; (**B**) GO enrichment analysis of genes from cluster 2; (**C**) gene co-expression patterns of cluster 254; (**D**) GO enrichment analysis of genes from cluster 254; (**E**) gene co-expression patterns of cluster 7; (**F**) GO enrichment analysis of genes from cluster 7; (**G**) gene co-expression patterns of cluster 58; (**H**) GO enrichment analysis of genes from cluster 58; (**A**,**C**,**E**,**G**) four clusters of gene co-expression patterns. The *y*-axis is the value of log_2_ measured by normalized gene expression.

**Figure 5 ijms-23-13986-f005:**
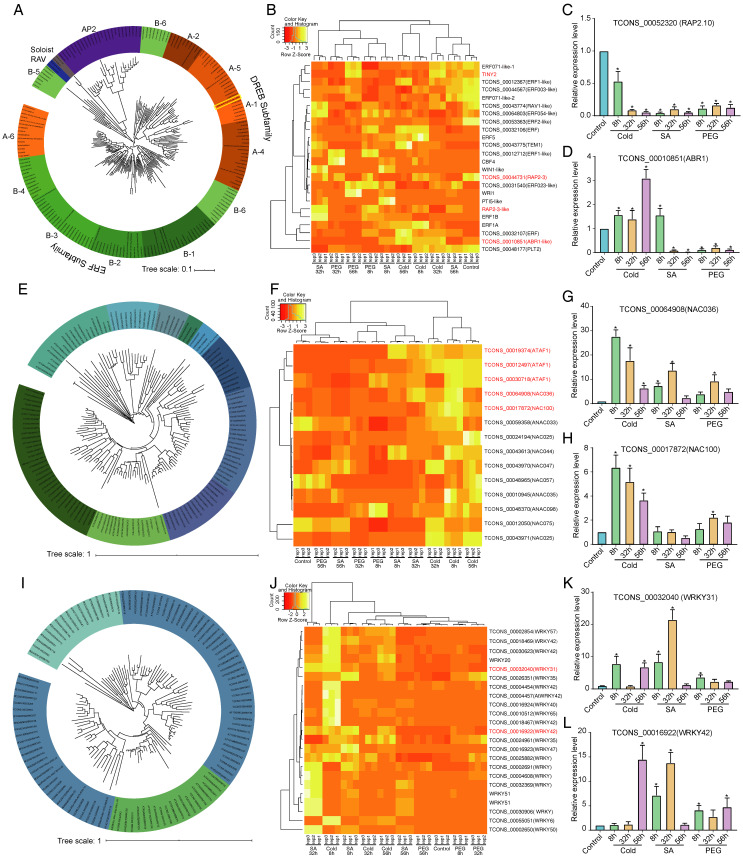
Systematic evolution analysis and differential expression of TFs under stress treatment. (**A**–**D**) evolution tree of AP2 TF family (**A**); heat map of differentially expressed AP2 family genes (**B**); and RT-qPCR verification of *RAP2.10* and *ABR1* genes of *C. lanceolata* (**C**,**D**); (**E**–**H**) evolution tree of NAC TF family (**E**); heat map of differentially expressed NAC family genes (**F**); and RT-qPCR verification of the *NAC036* and *NAC100* genes of *C. lanceolata* (**G**,**H**). (**I**–**L**) evolution tree of WRKY TF family (**I**); heat map of differentially expressed WRKY family genes (**J**); and RT-qPCR verification of the *WRKY31* and *WRKY42* genes of *C. lanceolata* (**K**,**L**). We performed three biological repeats in the RT-qPCR (standard deviation was used in the evaluation of error bar). Asterisks indicate significant differences between control and treatments at different times (*, *p* < 0.05, *t*-test).

**Figure 6 ijms-23-13986-f006:**
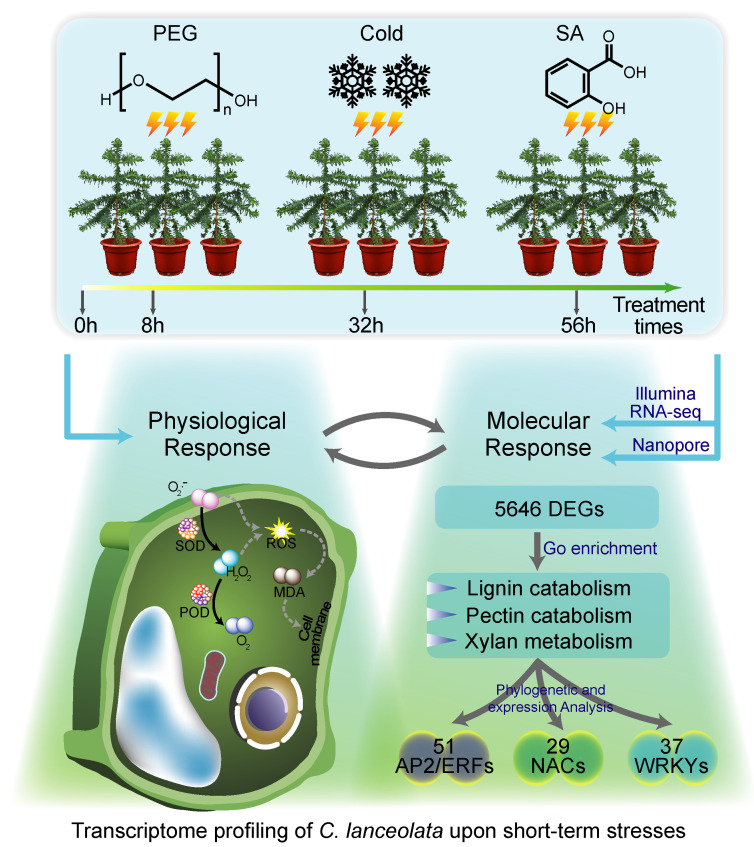
Model for stress responses in *C. lanceolata*.

## Data Availability

The accession numbers for the raw FAST5 file (nanopore DRS) and the FASTQ file (RNA-seq) are SRR12003701 and SRP266511, respectively. The RNA-Seq data analysis pipeline including Linux codes is available at https://github.com/GuInNGS/ChineseFir-different-srtess-RNA-seq (accessed on 19 October 2022).

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
