# Peer review of "Transcriptome Profiling of Stem-Differentiating Xylem in Response to Abiotic Stresses Based on Hybrid Sequencing in Cunninghamia lanceolata"

_ijms, 2022, doi:10.3390/ijms232213986_

Round 1

Reviewer 1 Report

Dear Authors, 

you can find all comments within attached the pdf file.

Author Response

Dear reviews,

We would like to express our gratitude for your handling of our manuscript during the review process. We thank you and other reviewers for their constructive comments on our manuscript titled “Transcriptome profiling of stem-differentiating xylem in re-sponse to abiotic stresses based on hybrid sequencing in Cunninghamia lanceolata”.

We took these comments to heart, and we believe they helped us improve our manuscript substantially. Based on the suggestions, we have made extensive revisions and addressed all reviewer questions and comments in the revised manuscript. We also used MDPI English editing services to polish the language. The changes to our manuscript are also highlighted within the document.

Review1

The suggestion from review1 was marked in the PDF manuscript.

Response to review1: Thank you; we appreciate your evaluation, which greatly improve our manuscript. Based on these suggestions, we have revised our manuscript. According to your suggestion, we revised the language and some sections, which did not clearly descript or were hard to understand. Moreover, we added the hypothesis in the introduction section. Finally, we also added sample information in the method section. We also used MDPI English editing services to polish the language. You can find this revision in the revised manuscript, which has been highlighted. We hope that you will find the revised version of our manuscript suitable for publication in the IJMS.

Best regards,

Professor, Lianfeng Gu

Fujian Agriculture and Forestry University, Fuzhou 350002, China

Tel: 86-591-8390305    Fax: 86-591-88250137

Reviewer 2 Report

Comments.

1)    Avoid using abbreviations and acronyms in the abstract.

2)    Does protein content come under physiological trait? Please re-check if needed.

3)    What is the basis of selecting the treatment time points, 0h, 8h, 32h and 56h? if needed elaborate in materials and methods section

4)    Did author determined RIN value of RNA before proceeding to sequencing? If so, please provide such information. 

5)    How author constructed the RNA-Seq libraries? Which kit or method used? What kind of adaptors used; such information is missing from methodology. Authors need to write such information (Materials &Methods :4.3)

6)    Full forms of RSEM and edgeR? (Materials &Methods :4.3)

7)    Please provide the RNA-Seq data analysis pipeline including Linux codes as a supplementary file.

8)    Co-expression analysis methodology is missing, please write in details how this method applied for this time-series analysis (Materials &Methods :4.3)

9)    Section-Materials & Methods :4.3 need to be divided into two or three parts i.e., sequencing and data analysis, make it separate

10) Why only few families of transcription factors were selected for RT-PCR? 

11) What is input for co-expression cluster analysis? TPM or FPKM? How did author log-transformed this?

12) The whole manuscript needs revision for language and grammar. 

Author Response

Dear reviews,

We would like to express our gratitude for your handling of our manuscript during the review process. We thank you and the other reviewer for their constructive comments on our manuscript titled “Transcriptome profiling of stem-differentiating xylem in re-sponse to abiotic stresses based on hybrid sequencing in Cunninghamia lanceolata”.

We took these comments to heart, and we believe they helped us improve our manuscript substantially. Based on the suggestions, we have made extensive revisions and addressed all reviewer questions and comments in the revised manuscript. We also used MDPI English editing services to polish the language. The changes to our manuscript are also highlighted within the document.

Review2

1)    Avoid using abbreviations and acronyms in the abstract.

Response: We apologize for the abbreviation and acronyms in the submitted manuscript. We corrected this issue in the revised manuscript.

2)    Does protein content come under physiological trait? Please re-check if needed.

Response: We are sorry that we did not describe our results more clearly in the first submitted version of our work. In the revised manuscript, we added following information: We took the protein content changes as physiological traits since the protein content was used in the determination of SOD and POD activity as described in the total superoxide dismutase (T-SOD) assay kit (hydroxylamine method) (Nanjing Jiancheng, A001-1) and the peroxidase assay kit (A084-3, Plant).

3)    What is the basis of selecting the treatment time points, 0h, 8h, 32h and 56h? if needed elaborate in materials and methods section

Response: We are sorry that we did not describe our results more clearly in the first submitted version of our work. In the revised manuscript, we added following information: In order to avoid the effect of circadian rhythms on gene expression, we collected materials at 24h intervals.

4)    Did author determined RIN value of RNA before proceeding to sequencing? If so, please provide such information. 

Response: We apologize for the missed RIN infromation in the submitted manuscript. We added “with RNA integrity number (RIN) values higher than 8” in the revised manuscript to make it clearly

5)    How author constructed the RNA-Seq libraries? Which kit or method used? What kind of adaptors used; such information is missing from methodology. Authors need to write such information (Materials &Methods :4.3)

Response: We apologize for the missed libraries construction infromation. In the revised manuscript, we added following information: The RNA-Seq was constructed using the deoxyuridine triphosphate (dUTP) method using Illumina Adapter Sequences (5' AATGATACGGCGACCACCGA GATCTACACIIIIIIIIAC ACTCTTTCCCTACACGACGC TCTTCCGATCT (-) AGATCGGAAGAGCACACGTC TGAACTCCAGTCACIIIIII IIATCTCGTATGCCGTCTTCTG CTTG 3') and the Illumina TruSeq Stranded mRNA kits.

6)    Full forms of RSEM and edgeR? (Materials &Methods :4.3)

Response: Thank you for this suggestion. We added the full form for RSEM and edgeR: Then, we used RNA-Seq by Expectation Maximization (RSEM) [70] and empirical analysis of DGE in R (edgeR) [71] to calculate the gene expression and the differentially expressed genes under the cold, PEG, and SA stress treatments.

7)    Please provide the RNA-Seq data analysis pipeline including Linux codes as a supplementary file.

Response: We totally agree with review’s suggest. In the revised manuscript, we added following information: The RNA-Seq data analysis pipeline including Linux codes is available at https://github.com/GuInNGS/ChineseFir-different-srtess-RNA-seq.

8)    Co-expression analysis methodology is missing, please write in details how this method applied for this time-series analysis (Materials &Methods :4.3)

Response: Thanks for your suggestion. We added following information into the revised manuscript: DPGP (https://github.com/PrincetonUniversity/DP_GP_cluster) performs very well in clustering gene expression time-series data using an infinite Gaussian process mixture model. After we obtained the DEGs, we extracted the FPKM values of the DEGs. Then, we used the FPKM values as inputs for the DPGP to cluster with the default parameters so as to identify similar expression disjoint clusters.

9)    Section-Materials & Methods :4.3 need to be divided into two or three parts i.e., sequencing and data analysis, make it separate

Response: We appreciate this comment. We have divided originated Section-Materials & Methods :4.3 into two parts: libraries construction and sequencing (4.3) and data analysis (4.4).

10) Why only few families of transcription factors were selected for RT-PCR? 

Response: This study included multiple transcription factors. It was difficult to include all the families for RT-PCR. Thus we only selected represented  families of transcription factors to save cost.

11) What is input for co-expression cluster analysis? TPM or FPKM? How did author log-transformed this?

Response: We used FPKM as input for DPGP to do co-expression cluster analysis. Then the DPGP will be transformed into a standardized log2 fold change in expression.

12) The whole manuscript needs revision for language and grammar. 

Response: We also used MDPI English editing services to polish the language.

We hope that you will find the revised version of our manuscript suitable for publication in the IJMS.

Best regards,

Professor, Lianfeng Gu

Fujian Agriculture and Forestry University, Fuzhou 350002, China

Tel: 86-591-8390305    Fax: 86-591-88250137

Round 2

Reviewer 1 Report

No additional comments

Reviewer 2 Report

I see no improvement in the manuscript. Methods for co-expression analysis are not presented accurately. 

The methodology used for experiments such as RNA sequencing is not properly drafted. Only a couple of sentences added. 

An English language edition is still required.